# Hyper-Relational Knowledge Representation Learning with Multi-Hypergraph Disentanglement

Submission Id: 697*

## Abstract

Hyper-relational knowledge graphs (HKGs) extend the traditional triplet-based knowledge graph by adding qualifiers to the relationships, making HKGs particularly useful for tasks that require more profound understanding and inference from relationships between entities. However, existing hyper-relational knowledge representation learning methods (HKRL) focus on direct neighbourhood information of entities only by neglecting the relational similarity of the main triple in hyper-relational facts and the attribute details in the qualifiers. In addition, few works extract common and private information across multiple views to minimize noise and interference. This paper proposes a multi-hypergraph disentanglement method for HKRL to address the above issues. Specifically, we first construct four hypergraphs to mine and utilise the inherent structure information of HKGs, and then propose to extract common representations among hypergraphs and private representations within individual hypergraphs to mine the semantic information and the task-relevant information, respectively. Experiment results on four real datasets demonstrate the effectiveness of the proposed method compared to SOTA methods in link prediction tasks on HKGs. Source code is available at the URL: https://anonymous.4open.science/r/MHD.

## CCS Concepts

• **Computing methodologies → Knowledge representation and reasoning**.

## Keywords

Hyper-relational knowledge graph; Hypergraph; Multi-view; Information disentangled

**ACM Reference Format:**

Anonymous Author(s). 2025. Hyper-Relational Knowledge Representation Learning with Multi-Hypergraph Disentanglement. In *Proceedings of Proceedings of the ACM Web Conference 2025 (Conference WWW '25)*. ACM, New York, NY, USA, 11 pages. https://doi.org/XXXXXXX.XXXXXXX

## 1 Introduction

Knowledge graphs (KGs) have profoundly permeated the Web domain, acting as a powerful force that drives the advancement of semantic search [35], question answering [11], and personalized

recommendation [24, 34]. The traditional KG is a graph structure composed of nodes (representing entities) and edges (representing relations), and is often described by triples (*i.e.*, $(s, r, o)$) [8, 13]. However, triples exhibit inherent limitations in describing knowledge, namely, their formal simplicity inevitably leads to ambiguity [4]. Hyper-relational knowledge graphs (HKGs) have emerged as an alternative by expanding the traditional triple structure to the hyper-relational fact (H-Fact) [3, 12], *i.e.*, incorporating qualifiers as attribute extension elements. For example, HKGs specifies "*(Cinderella, voice actor, Luis van Rooten)*" in traditional knowledge graph as "*{(Cinderella, voice actor, Luis van Rooten), character role: The King and Grand Duke}*", to offer greater flexibility and expressiveness in modelling real-world knowledge. Recently, hyper-relational knowledge representation learning (HKRL) has become a hot spot in academia and industry with the goal of developing effective techniques for learning the representations of entities and relations within HKGs, so that these representations enable intelligent systems to understand and utilise the H-Fact in HKGs [2, 15].

Existing HKRL methods can be broadly divided into two categories based on their utilisation of neighbourhood information, *i.e.*, intra-hyperrelational representation methods and neighbourhood-aware representation methods. The intra-hyperrelational representation method focuses on the structure and association information within individual H-Facts [7, 25, 28]. For example, HINGE [20] employs two graph convolutional neural networks to capture the structural information of the main triple and the correlation between the triple and its attribute-value pairs. GRAN [26] models each H-Fact as a heterogeneous graph to represent the H-Fact's internal structure and applies graph learning to capture entity relation associations within the H-Fact. However, in practical scenarios, H-Facts are often interconnected rather than isolated. Ignoring the relationships between H-Facts and the overall structure of HKGs limits the potential of these models. The neighbourhood-aware representation method addresses such limitations by incorporating the neighbourhood information of the H-fact in the HKG structure to exploit the connectivity of H-Facts. For example, HAHE [16] constructs a global-level hypergraph to represent the connectivity of entities and uses hypergraph dual-attention layers to learn structural information in the HKG. HyperFormer [10] introduces an entity neighbour aggregator to integrate the neighbourhood information of entities within an H-Fact.

Previous HKRL methods have demonstrated their effectiveness, but they still have limitations to be addressed. First, existing HKRL methods consider neighbourhood information of entities only by neglecting either the similarity of relations in the main triple or the identity of attributes in the qualifiers, so that they difficult extract semantic similarity in the HKG. For example, given three H-Facts as follows, *i.e.*, $\mathcal{F}_1$ = *{(Cinderella, voice actor, Luis van Rooten), {character role: The King and Grand Duke, publication time: 1950}}*, $\mathcal{F}_2$ =

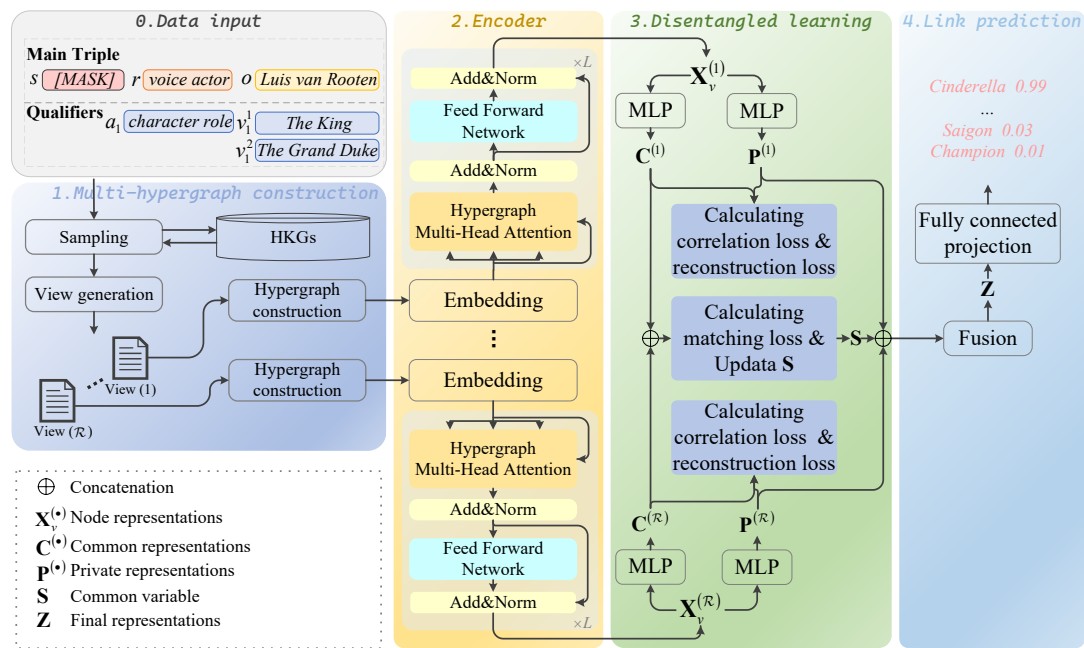

**Figure 1: The flowchart of the proposed MHD consists of four components, *i.e.,* multi-hypergraph construction, hypergraph encoder, disentangled learning, and link prediction. Specifically, the multi-hypergraph construction module first constructs four types of neighbourhood views, *i.e.,* subject view, relation view, object view, and qualifier view, based on the structural characteristics of H-Facts, and then transforms every view to a hypergraph structure. The hypergraph encoder module employs a transformer-based hypergraph encoder to extract both node and hyperedge representations from every hypergraph. The disentangled learning module disentangles the representations among different graphs into common and private representations. Finally, the fusion representation is used to conduct link prediction in the link prediction module.**

*{(Cinderella, voice actor, Ilene Woods), {character role: Cinderella}}*, and $\mathcal{F}_3$ = *{(Cinderella, directed by, Clyde Geronimi), {publication time: 1950}}*, $\mathcal{F}_1$ and $\mathcal{F}_2$ describe the voice actors for characters in the movie "*Cinderella*", sharing the relation "*voice actor*". This similarity of relation implies that the two H-Facts contain similar semantic information. In addition, both $\mathcal{F}_1$ and $\mathcal{F}_3$ have the qualifier "*publication time: 1950*", indicating that their events were published in 1950. Therefore, exploring relation similarity and the qualifier identity is crucial to mining comprehensive semantic patterns and improving hyper-relational knowledge representation.

Second, existing HKRL methods mine the neighbourhood information of H-Facts by ignoring HKG's diversity and thus decreasing the quality of its representation. For example, HAHE [16] constructs a hypergraph to represent the connectivity of entities by ignoring the semantic information between relations. Actually, the connections and interactions between entities help to understand their roles and positions within H-Facts from the perspective of entities, while the similarity between relations helps to understand their structural semantics from the perspective of relations. Therefore, combining different perspectives to construct multiple hypergraphs can capture the diversity of the HKG to further enrich the representation of H-Facts. However, few study focused on this.

In this paper, we propose a Multi-Hypergraph Disentanglement (MHD) method for hyper-relational knowledge representation learning, as shown in Figure 1, to address the above issues. Specifically,

based on the structural characteristics of H-Facts, we first sample four neighbourhood perspectives from HKGs and then transform these perspectives into different hypergraph structures, *i.e.,* the subject hypergraph, the relation hypergraph, the object hypergraph, and the qualifier hypergraph. As a result, the relation hypergraph and the qualifier hypergraph, respectively, are designed to explore the similarity of relations and identity of qualifiers, thus addressing the first issue. Moreover, our method constructs multiple hypergraphs to consider the diversity of HKGs and address the second issue. Considering that every hypergraph contains both the semantic information of H-Fact and the structural information of HKG, we disentangle the representations of nodes into common representation and private representation, aiming at using common representation to extract the semantic information and use private representation to mine the task-relevant information (*e.g.,* structure information of HKGs).

Compared with previous HKRL methods, our contributions are summarized as follows:

- We construct a multi-hypergraph for HKRL to mine the relation similarity and the qualifier identity in HKGs, as well as comprehensively utilise the neighbourhood information of H-Facts.
- We propose to disentangle different kinds of representations for nodes by mining semantic information of H-Facts and task-relevant information of the HKRL. To the best of our

knowledge, this is the first attempt to decouple common and private representations for HKRL.

## 2 Methodology

**Notations.** Denoting $\mathcal{H}_{KG} = \{\mathcal{F}_1, \cdots, \mathcal{F}_U\}$ a hyper-relational knowledge graph, $\mathcal{F} = \{\mathcal{T}, \mathcal{Q}\}$ is the hyper-relational fact (H-Fact), where $\mathcal{T} = \{(s, r, o)|s, o \in E, r \in R\}$ is a main triple, $\mathcal{Q} = \{\{a_i : \{v_i^j\}_{j=1}^{n_i}\}_{i=1}^{m}|v_i^j \in E, a_i \in R\}$ is the set of qualifiers, the attribute $a_i$ and the value set $\{v_i^j\}_{j=1}^{n_i}$ form $i$-th qualifier. Denoting $E$ the set of entities, $R$ the set of relations, $m$ and $n$ the number of qualifiers and the quantity of values, respectively, the goal of the proposed MHD method is to learn the representation $\mathbf{Z} \in \mathbb{R}^{|\mathcal{F}| \times d}$ of H-Fact $\mathcal{F} \in \mathcal{H}_{KG}$, where $|\mathcal{F}|$ is the number of elements in the H-Fact $\mathcal{F}$, and $d$ is the dimension of representations.

### 2.1 Motivation

Given a H-Fact $\mathcal{F} = \{(s, r, o), \{a_i : \{v_i^j\}_{j=1}^{n_i}\}_{i=1}^{m}\}$, existing methods first extract one-hop neighbourhood information of entities (*i.e.*, $s$, $o$ and $v$) to construct a hypergraph for learning the representation of entities within the H-Fact, and then employ a local-level encoder to learn the semantic information of H-Facts [10]. Obviously, the hypergraph construction is the most important step. Previous methods often involve two steps to construct the hypergraph, *i.e.*, extracting one-hop neighbourhood of entities and constructing a hypergraph of entities. Specifically, denoting the one-hop neighbourhood of entities $s$, $o$, and every attribute value $v_i^j$, respectively, $\mathcal{N}(s) = \{\mathcal{F}' \in \mathcal{H}_{KG} \mid s \in (\mathcal{F}' \cap \mathcal{F})\}$, $\mathcal{N}(o) = \{\mathcal{F}' \in \mathcal{H}_{KG} \mid o \in (\mathcal{F}' \cap \mathcal{F})\}$, and $\mathcal{N}(v_i^j) = \{\mathcal{F}' \in \mathcal{H}_{KG} \mid v_i^j \in (\mathcal{F}' \cap \mathcal{F})\}$. The neighbourhood of H-Fact $\mathcal{F}$ is denoted by $\mathcal{N}(\mathcal{F}) = \{\mathcal{N}(s) \cup \mathcal{N}(o) \cup \mathcal{N}(v_i^j)\}$. The hypergraph is denoted by $\mathcal{G} = (\mathcal{V}, \mathcal{E}, \mathbf{H})$. The node set $\mathcal{V}$ consists of all subjects, objects, and values from the auxiliary attributes, *i.e.*, $\mathcal{V} = \{s, o\} \cup \{v_i^j \mid i = 1, \cdots, m; j = 1, \cdots, n_i\} \cup (\bigcup_{\mathcal{F}' \in \mathcal{N}(\mathcal{F})} \{s', o'\}) \cup (\bigcup_{\mathcal{F}' \in \mathcal{N}(\mathcal{F})} \{v_i^{j'} \mid i = 1, \cdots, m'; j = 1, \cdots, n_i'\})$. The hyperedge set $\mathcal{E}$ is formed by all H-Facts, where each hyperedge connects an entity pairs $(s, o)$ and its auxiliary attribute values $v_i^j$, *i.e.*, $\mathcal{E} = \{e_{\mathcal{F}'} \mid \mathcal{F}' \in (\mathcal{N}(\mathcal{F}) \cup \{\mathcal{F}\})\}$. The element of the indicator matrix $\mathbf{H} \in \mathbb{R}^{|\mathcal{V}| \times |\mathcal{E}|}$ is defined by:

$$h_{ij} = \begin{cases} 1, & \text{if } v_i \in e_j; \ v_i \in \mathcal{V}; \text{ and } e_j \in \mathcal{E}, \\ 0, & \text{if } v_i \notin e_j; \ v_i \in \mathcal{V}; \text{ and } e_j \in \mathcal{E}. \end{cases} \quad (1)$$

After constructing the hypergraph $\mathcal{G} = (\mathcal{V}, \mathcal{E}, \mathbf{H})$ from H-Facts, existing methods often use an hypergraph encoder to extract representations for the nodes in the hypergraph, *i.e.*,

$$\mathbf{X}_v = f_h\left(\mathbf{X}_v^0, \mathbf{H}\right). \quad (2)$$

where $\mathbf{X}_v^0 \in \mathbb{R}^{|\mathcal{V}| \times d_0}$ is the initialised node representation, $d_0$ is the initial node dimension, and $\mathbf{X}_v \in \mathbb{R}^{|\mathcal{V}| \times d}$ is the updated node representation. As a result, node representations learn the structural information from the HKG, so a local-level encoder is then used to learn the semantic information of H-Facts, *i.e.*,

$$\mathbf{Z} = f_l\left(\mathbf{X}_v, \mathcal{F}\right) \quad (3)$$

where $\mathbf{Z} \in \mathbb{R}^{|\mathcal{F}| \times d}$ is the updated representation of the H-Fact $\mathcal{F}$.

However, previous methods still have limitations. First, the existing methods mix all types of neighbouring entities into a single

hypergraph, which may lead to information interference among different types of neighbours. For example, the neighbours of the subject entity may convey context relevant to the subject, while the neighbours of the object entity provide context relevant to the object. Combining them may dilute or confuse key features, thus negatively impacting the representation learning of entities. Second, the existing methods only consider the neighbourhood feature of the entities (*i.e.*, $s$, $o$ and $v$) to extract the structural information of HKGs, ignoring the importance of relation $r$ in the main triple and attribute $a$ in the qualifiers.

### 2.2 Multi-hypergraph construction

To address the above limitation, we construct four different neighbourhood hypergraphs based on the structural characteristics of the H-Fact to mine comprehensive semantic information in HKGs. Specifically, we divide the construction process of every hypergraph into two steps, *i.e.*, the neighbourhood view generation and the hypergraph construction. We list the details as follows.

(1) We sample four neighbourhood views based on the type of element in H-Facts, *i.e.*,

- generating the one-hop neighbourhood $\mathcal{N}(s)$ of the subject entity $s$ as:

$$\mathcal{N}(s) = \{\mathcal{F}' \in \mathcal{H}_{KG} \mid s \in (\mathcal{F}' \cap \mathcal{F})\}. \quad (4)$$

- generating the one-hop neighbourhood $\mathcal{N}(r)$ of the relation $r$ as:

$$\mathcal{N}(r) = \{\mathcal{F}' \in \mathcal{H}_{KG} \mid r \in (\mathcal{F}' \cap \mathcal{F})\}. \quad (5)$$

- generating the one-hop neighbourhood $\mathcal{N}(o)$ of the object entity $o$ as:

$$\mathcal{N}(o) = \{\mathcal{F}' \in \mathcal{H}_{KG} \mid o \in (\mathcal{F}' \cap \mathcal{F})\}. \quad (6)$$

- generating the one-hop neighbourhood $\mathcal{N}(av)$ for each qualifier $\{a_i : \{v_i^j\}\}$ as:

$$\mathcal{N}(av) = \{\mathcal{F}' \in \mathcal{H}_{KG} \mid \{a_i, v_i^j\} \subset (\mathcal{F}' \cap \mathcal{F})\}. \quad (7)$$

(2) We construct a hypergraph for every view to capture higher-order relationships among H-Facts for node representation learning, *i.e.*, neighbourhood hypergraphs $\mathcal{G}^{(s)}$, $\mathcal{G}^{(r)}$, $\mathcal{G}^{(o)}$, and $\mathcal{G}^{(av)}$. For brevity, we illustrate the construction process of the hypergraph $\gamma$ ( *i.e.*, $\mathcal{G}^{(\gamma)} = (\mathcal{V}^{(\gamma)}, \mathcal{E}^{(\gamma)}, \mathbf{H}^{(\gamma)})$, where $\gamma \in \{s, r, o, av\}$) as follows.

- The node set $\mathcal{V}^{(\gamma)}$ consists of all elements, including the subject, the relation, the object, the attribute and values and is denoted by:

$$\mathcal{V}^{(\gamma)} = \{s, r, o\} \cup \left\{a_i, v_i^j \mid i = 1, \cdots, m; j = 1, \cdots, n_i\right\}$$
$$\cup \left(\bigcup_{\mathcal{F}' \in \mathcal{N}(\gamma)} \{s', r', o'\}\right) \quad (8)$$
$$\cup \left(\bigcup_{\mathcal{F}' \in \mathcal{N}(\gamma)} \left\{a_i', v_i^{j'} \mid i = 1, \cdots, m'; j = 1, \cdots, n_i'\right\}\right).$$

- The hyperedge set $\mathcal{E}^{(\gamma)}$ is formed by each H-Fact, where each hyperedge connects a triple $\{s, r, o\}$ and its qualifiers

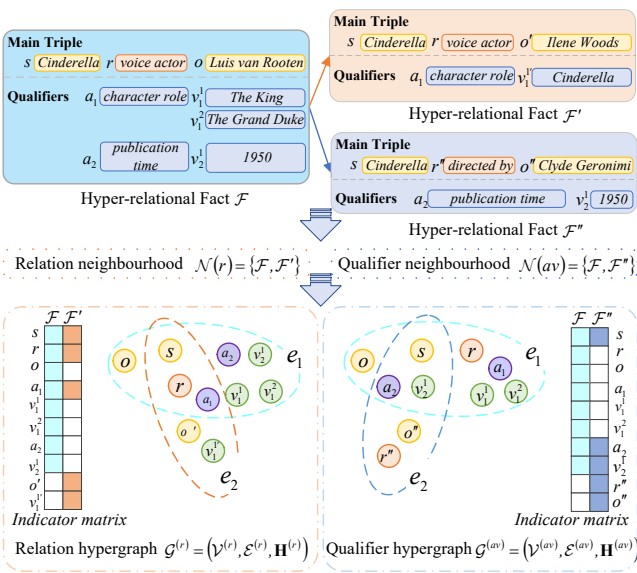

**Figure 2: Illustration of the hypergraph construction of the relation hypergraph $\mathcal{G}^{(r)}$ (left) and the qualifier hypergraph $\mathcal{G}^{(av)}$ (right), where each hypergraph contains one neighbouring H-Fact of the H-Fact $\mathcal{F}$.**

$$\left\{ a_i, v_i^j \mid i = 1, \cdots, m; j = 1, \cdots, n_i \right\} \text{ and is denoted by:}$$

$$\mathcal{E}^{(\gamma)} = \left\{ e_{\mathcal{F}'} \mid \mathcal{F}' \in (\mathcal{N}(\gamma) \cup \{\mathcal{F}\}) \right\}. \tag{9}$$

- The indicator matrix $\mathbf{H}^{(\gamma)} \in \mathbb{R}^{|\mathcal{V}^{(\gamma)}| \times |\mathcal{E}^{(\gamma)}|}$ is defined by:

$$h_{ij} = \begin{cases} 1, & \text{if } v_i \in e_j; v_i \in \mathcal{V}^{(\gamma)}; \text{ and } e_j \in \mathcal{E}^{(\gamma)}, \\ 0, & \text{if } v_i \notin e_j; v_i \in \mathcal{V}^{(\gamma)}; \text{ and } e_j \in \mathcal{E}^{(\gamma)}. \end{cases} \tag{10}$$

Based on the above two steps, we obtain four hypergraphs, *i.e.*, the subject hypergraph $\mathcal{G}^{(s)} = (\mathcal{V}^{(s)}, \mathcal{E}^{(s)}, \mathbf{H}^{(s)})$, the relation hypergraph $\mathcal{G}^{(r)} = (\mathcal{V}^{(r)}, \mathcal{E}^{(r)}, \mathbf{H}^{(r)})$, the object hypergraph $\mathcal{G}^{(o)} = (\mathcal{V}^{(o)}, \mathcal{E}^{(o)}, \mathbf{H}^{(o)})$, and the qualifier hypergraph $\mathcal{G}^{(av)} = (\mathcal{V}^{(av)}, \mathcal{E}^{(av)}, \mathbf{H}^{(av)})$. Figure 2 lists a visual example to show the process of the hypergraph construction.

Compared to previous methods, our multi-hypergraph construction shows at least two advantages. First, we explore both the relation similarity and the qualifier identity in HKGs to comprehensively utilise the neighbourhood information of H-Facts. Second, multi-hypergraph construction can capture different perspectives of H-Facts. For example, a relation hypergraph can be constructed to describe the semantic similarity among H-Facts, while a qualifier hypergraph can represent the attribute identity. Each hypergraph focuses on a specific information perspective, providing the model with rich and comprehensive knowledge representation.

## 2.3 Hypergraph encoding

In hypergraph representation learning, one of the key challenges is extracting higher-order feature information. Current approaches often rely on convolutional neural networks or attention networks, such as HyperGCN [33], HGNN [5], and TDHNN [38], to offer new

insights into data with complex relationships. However, in hyper-relational knowledge graphs, if the number of nodes and edges is immense, the connections between two nodes tend to be sparse. This sparsity hinders the model to effectively capture the intricate relationships between two nodes during the training process, ultimately impacting the quality of the learned representations. To address these issues, we propose a hypergraph transformer encoder.

The proposed encoder stacks $L$ layers and every layer has a hypergraph multi-head self-attention and a feed-forward network [22], whose connection is residual connections [9] and layer normalization operation [32]. For simplicity, we use the symbol $\mathcal{G} = (\mathcal{V}, \mathcal{E}, \mathbf{H})$ to denote the hypergraph in this section.

Given the hypergraph $\mathcal{G} = (\mathcal{V}, \mathcal{E}, \mathbf{H})$, we first employ an embedding layer (*i.e.*, word embedding [17] or random initialization [38]) to initial node representations $\mathbf{X}_v^0 \in \mathbb{R}^{|\mathcal{V}| \times d}$ and hyperedge representations $\mathbf{X}_e^0 \in \mathbb{R}^{|\mathcal{E}| \times d}$, where $|\mathcal{V}|$, $|\mathcal{E}|$, and $d$, respectively, are the number of nodes, hyperedges of the hypergraph $\mathcal{G}$, the dimension of the representation. After that, we feed the initialised $\mathbf{X}_v^0$ and $\mathbf{X}_e^0$, as well as the incidence matrix $\mathbf{H} \in \mathbb{R}^{|\mathcal{V}| \times |\mathcal{E}|}$ into our hypergraph encoder with $L$ layers to update the representation for the nodes and hyperedges. The hypergraph multi-head self-attention has two steps, *i.e.*, nodes to hyperedges and hyperedges to nodes.

(1) Node features are aggregated into hyperedge representations to capture the high-order information within H-Fact, *i.e.*,

$$\tilde{\mathbf{X}}_e^l = Softmax \left( \frac{(\mathbf{W}_e^Q \mathbf{X}_e^{l-1}) \cdot (\mathbf{W}_v^K \mathbf{X}_v^{l-1})^\top}{\sqrt{d_z}} \odot \mathbf{H} \right) \cdot (\mathbf{W}_v^V \mathbf{X}_v^{l-1}) \tag{11}$$

$$\mathbf{X}_e^l = MLP \left( \tilde{\mathbf{X}}_e^l \oplus \mathbf{X}_e^{l-1} \right) \tag{12}$$

where $\mathbf{W}_e^Q$, $\mathbf{W}_v^K$, and $\mathbf{W}_v^V$ are the learnable weights, $\odot$ represents the Hadamard multiplication, $\oplus$ means the concatenation operation, $d_z = \frac{d}{T}$ is the dimension of the heads in all layers, and $T$ is the number of the heads.

(2) We aggregate the hyperedge representations into the node representations to achieve high-order information fusion, *i.e.*,

$$\tilde{\mathbf{X}}_v^l = Softmax \left( \frac{(\mathbf{W}_v^Q \mathbf{X}_v^{l-1}) \cdot (\mathbf{W}_e^K \mathbf{X}_e^l)^\top}{\sqrt{d_z}} \odot \mathbf{H} \right) \cdot (\mathbf{W}_e^V \mathbf{X}_e^l) \tag{13}$$

where $\mathbf{W}_v^Q$, $\mathbf{W}_e^K$, and $\mathbf{W}_e^V$ are the learnable weights.

After the above two steps, we employ feed-forward networks $f_{FFN}(\cdot)$ to capture complex patterns through non-linear transformations as well as employ layer normalization $f_{LN}(\cdot)$ to ensure training stability, fast convergence, and effective gradient flow, *i.e.*,

$$\dot{\mathbf{X}}_v^l = f_{FFN} \left( f_{LN}(\tilde{\mathbf{X}}_v^l + \mathbf{X}_v^{l-1}) \right) \tag{14}$$

$$\mathbf{X}_v^l = f_{LN} \left( \dot{\mathbf{X}}_v^l + f_{LN}(\tilde{\mathbf{X}}_v^l + \mathbf{X}_v^{l-1}) \right). \tag{15}$$

Compared to previous hypergraph representation learning methods, our method has at least two advantages as follows. First, it can capture the relationships between any two nodes in the hypergraph. This global interaction helps alleviate the issue of data sparsity. Second, through the self-attention mechanism, our model can learn the intrinsic connections and importance differences between nodes, thereby mitigating the impact of noise and missing data on model performance.

## 2.4 Disentangled learning

Every hypergraph contains the similar content to describe the same H-Fact (*i.e.,* common information), as well as contains the information different from other hypergraphs (*i.e.,* private information). For example, every hypergraph contains the H-Fact $\mathcal{F}$ (*i.e.,* $\mathcal{F} \in \{\mathcal{E}^{(s)} \cap \mathcal{E}^{(r)} \cap \mathcal{E}^{(o)} \cap \mathcal{E}^{(av)}\}$) and the node representation in different hypergraphs captures the same semantic information of the H-Fact. In addition, every hypergraph contains a different neighbour set (*i.e.,* $\{\mathcal{E}^{(s)} - \mathcal{F}\} \neq \{\mathcal{E}^{(r)} - \mathcal{F}\} \neq \{\mathcal{E}^{(o)} - \mathcal{F}\} \neq \{\mathcal{E}^{(av)} - \mathcal{F}\}$), which provides additional knowledge and context to the H-Fact, helping to enhance knowledge reasoning and link prediction tasks.

After the hypergraph encoder, the node representations of H-Fact $\mathcal{F}$ in each hypergraph simultaneously capture the semantic information within the H-Fact and the structural information of the HKG. However, semantic information is often intertwined with the structural information, forming complex entangled representations. As a result, fusing node representations of the H-Fact in multi-hypergraphs will lead to semantic information redundancy. Effectively extracting high-quality and reliable semantic information of H-Facts from these intertwined multi-hypergraphs has become a key challenge in current research [19, 30]. In addition, complete structure information in multi-hypergraph provides additional knowledge and context to the H-Fact, helping to enhance knowledge reasoning and link prediction tasks. Thus, extracting clean semantic information and complete structure information helps to improve the performance of HKRL.

To do this, we explore a multi-hypergraph disentangled representation learning to use common representation to extract semantic information as well as to use private representation to explore task-relevant information (*e.g.,* structure information of HKG) for the HKRL. Specifically, we extract common and private representations from the node representations of the H-Fact $\mathcal{F}$ by:

$$\begin{cases} \mathbf{C}^{(\gamma)} = f_c \left( \mathbf{X}_{\mathcal{F}}^{(\gamma)} \right) \\ \mathbf{P}^{(\gamma)} = f_p \left( \mathbf{X}_{\mathcal{F}}^{(\gamma)} \right) \end{cases} \tag{16}$$

where $\mathbf{X}_{\mathcal{F}}^{(\gamma)} \in \mathbb{R}^{|\mathcal{F}| \times d}$ is the node representations of the H-Fact $\mathcal{F}$ in the hypergraph $\gamma$, $|\mathcal{F}|$ is the number of nodes in the H-Fact $\mathcal{F}$. $\mathbf{C}^{(\cdot)} \in \mathbb{R}^{|\mathcal{F}| \times d_c}$ and $\mathbf{P}^{(\cdot)} \in \mathbb{R}^{|\mathcal{F}| \times d_p}$ are the common representation and private representation, respectively. $d_c$ and $d_p$ are the dimensions of common and private representation, respectively. $f_c(\cdot)$ and $f_p(\cdot)$ are two mapping functions (*i.e.,* multilayer perceptron [21]) with non-shared parameters.

In order to disentangle two kinds of information, we first extract a common variable matrix $\mathbf{S}$ from common representations in four hypergraphs by singular value decomposition, which is both orthogonal and zero-mean. Specifically, we introduce a matching loss to minimize the distinctions between the common variable matrix $\mathbf{S}$ and the common representations $\mathbf{C}^{(\gamma)}$ from different hypergraphs, encouraging high consistency among common representations of all hypergraphs, *i.e.,*

$$\mathcal{L}_{mat} = \frac{1}{|\mathcal{F}|} \sum_{\gamma=1}^{4} \sum_{i=1}^{|\mathcal{F}|} \left\| \mathbf{c}_i^{(\gamma)} - \mathbf{s}_i \right\|_2^2, s.t. \mathbf{SS}^\top = \mathbf{I}, \frac{1}{|\mathcal{F}|} \sum_{i=1}^{|\mathcal{F}|} \mathbf{s}_i = \mathbf{0} \tag{17}$$

where $\|\cdot\|_2$ represents the $L2$ norm.

Second, we decouple common and private representations to enforce their independence, achieving clean common information and complete private information. Specifically, we use the Pearson's correlation coefficient to calculate a correlation loss to achieve this goal, *i.e.,*

$$\mathcal{L}_{cor} = \sum_{\gamma=1}^{4} \frac{\left| Cov \left( f_\phi^{(\gamma)}(\mathbf{C}^{(\gamma)}), f_\psi^{(\gamma)}(\mathbf{P}^{(\gamma)}) \right) \right|}{\sqrt{Var \left( f_\phi^{(\gamma)}(\mathbf{C}^{(\gamma)}) \right)} \sqrt{Var \left( f_\psi^{(\gamma)}(\mathbf{P}^{(\gamma)}) \right)}} \tag{18}$$

where $f_\phi^{(\gamma)}(\cdot)$ and $f_\psi^{(\gamma)}(\cdot)$ are measurable functions, $Cov(\cdot, \cdot)$ and $Var(\cdot)$ indicate covariance and variance operations, respectively. This correlation loss reinforces the statistical independence between common and private representations, ensuring they remain maximally separated and non-interfering in the feature space.

The learned common and private representations may result in trivial solutions. Existing methods usually adopt adversarial training or auto-encoders [18, 31]. However, these methods do not consider node features and hypergraph structure reconstruction simultaneously. To address these issues, we propose a reconstruction loss function, *i.e.,*

$$\mathcal{L}_{rec} = \sum_{\gamma=1}^{4} \left( \frac{\left\| \tilde{\mathbf{X}}_{\mathcal{F}}^{(\gamma)} - \mathbf{X}_{\mathcal{F}}^{(\gamma)} \right\|_F^2}{|\mathcal{F}|} + \frac{\left\| \mathbf{X}_v^{(\gamma)} \mathbf{X}_e^{(\gamma)^\top} - \mathbf{H}^{(\gamma)} \right\|_F^2}{|\mathcal{V}^{(\gamma)}| \times |\mathcal{E}^{(\gamma)}|} \right) \tag{19}$$

where $\tilde{\mathbf{X}}_{\mathcal{F}}^{(\cdot)} = MLP(\mathbf{C}^{(\cdot)} \oplus \mathbf{P}^{(\cdot)})$ is the reconstructed node representations obtained by concatenating $\mathbf{C}^{(\cdot)}$ and $\mathbf{P}^{(\cdot)}$ and feeding them into the reconstruction mapping function. The second term in the equation represents the reconstruction of the topology of hypergraph $\gamma$.

Compared to existing methods, our disentangled learning shows at least two advantages. First, it uses common representation to capture the semantic information and private representation to extract the structural information. This separation captures clearer semantic information about H-Fact, and thus improving HKRL's performance. Second, our method ensures that additional structural information from multiple hypergraphs provides a more comprehensive context to the H-Facts, thus supporting better knowledge reasoning and more accurate link prediction tasks.

## 2.5 Link prediction

After conducting the processes, including the multi-hypergraph construction, the hypergraph encoder, and disentangled representation learning, we obtain the common variable matrix $\mathbf{S}$ and private representations $\mathbf{P}$ for the H-Fact. Given a H-Fact with missing values, previous methods of link prediction fuse private representations from multiple views to a unified representation using summation or average pooling. However, either summation or average pooling simply applies weighted addition or averaging to representations from different perspectives, which ignores the differences between features from each perspective. If one perspective is critical while others are relatively noisy, average pooling will dilute the contribution of the necessary perspective. To address these issues, we employ an attention mechanism to automatically learn the importance of different hypergraphs and dynamically adjust

**Table 1: Evaluation of different models with mixed-percentage mixed-qualifier on the JF17K, WD50K, WikiPeople-, and WikiPeople datasets. Best scores are highlighted in bold, the second best scores are underlined, and '–' indicates the results are not reported in previous work.**

| Model | JF17K | | | WD50K | | | WikiPeople- | | | WikiPeople | | |
|---|---|---|---|---|---|---|---|---|---|---|---|---|
| | MRR | H@1 | H@10 | MRR | H@1 | H@10 | MRR | H@1 | H@10 | MRR | H@1 | H@10 |
| m-TransH | 0.206 | 0.206 | 0.462 | - | - | - | 0.063 | 0.063 | 0.300 | - | - | - |
| RAE | 0.215 | 0.215 | 0.466 | - | - | - | 0.059 | 0.059 | 0.306 | - | - | - |
| NaLP | 0.221 | 0.165 | 0.331 | 0.177 | 0.131 | 0.264 | 0.408 | 0.331 | 0.546 | 0.206 | 0.161 | 0.291 |
| HINGE | 0.449 | 0.361 | 0.624 | 0.243 | 0.176 | 0.377 | 0.445 | 0.359 | 0.577 | 0.318 | 0.238 | 0.457 |
| Transformer | 0.512 | 0.434 | 0.665 | 0.286 | 0.222 | 0.406 | 0.469 | 0.403 | 0.586 | 0.426 | 0.335 | 0.576 |
| STARE | 0.572 | 0.493 | 0.725 | 0.318 | 0.215 | 0.496 | 0.478 | 0.393 | 0.571 | 0.435 | 0.326 | 0.601 |
| ShrinkE | 0.565 | 0.489 | 0.711 | 0.210 | 0.170 | 0.285 | 0.421 | 0.357 | 0.607 | 0.385 | 0.286 | 0.571 |
| GRAN | 0.567 | 0.485 | 0.728 | 0.332 | 0.262 | 0.466 | 0.476 | 0.382 | 0.623 | 0.456 | 0.364 | 0.602 |
| NYLON | 0.466 | 0.374 | 0.651 | 0.290 | 0.225 | 0.413 | 0.459 | 0.358 | 0.624 | 0.383 | 0.298 | 0.519 |
| HyperFormer | 0.659 | 0.594 | **0.785** | 0.370 | 0.292 | 0.519 | 0.470 | 0.357 | **0.645** | 0.434 | 0.325 | 0.610 |
| HAHE | 0.598 | 0.518 | 0.758 | 0.349 | 0.273 | 0.493 | 0.480 | 0.397 | 0.618 | 0.465 | 0.382 | 0.599 |
| MHD (our) | **0.696** | **0.669** | 0.745 | **0.488** | **0.453** | **0.553** | **0.544** | **0.506** | 0.614 | **0.538** | **0.499** | **0.611** |

the contribution of each perspective, *i.e.,*

$$\mathbf{P} = f_{att}\left(\mathbf{P}^{(s)} \oplus \mathbf{P}^{(r)} \oplus \mathbf{P}^{(o)} \oplus \mathbf{P}^{(av)}\right) \quad (20)$$

where $f_{att}$ is a attention function (*i.e.,* MLP). We then get the final node representation $\mathbf{Z}$ by concatenating the common variable $\mathbf{S}$ and $\mathbf{P}$ of the above formula, *i.e.,*

$$\mathbf{Z} = f_z\left(\mathbf{S} \oplus \mathbf{P}\right) \quad (21)$$

After obtaining the final representation $\mathbf{Z}$, we predict the missing node $\mathbf{z}_{mask}$ in the H-Fact by:

$$\tilde{\mathbf{y}} = Softmax\left(f_{pre}(\mathbf{z}_{mask})\right) \quad (22)$$

where $f_{pre}$ is a prediction function (*i.e.,* MLP) and $\tilde{\mathbf{y}}$ is the probability distribution of the missing element.

Finally, we use the cross-entropy loss function [37] as the link prediction loss, *i.e.,*

$$\mathcal{L}_{link} = \frac{1}{\mathcal{S}}\sum_{i=1}^{\mathcal{S}} y_i \log \tilde{y}_i \quad (23)$$

where $y_i$ and $\tilde{y}_i$ are the $i$-th entry of the label $\mathbf{y}$ and the prediction $\tilde{\mathbf{y}}$, respectively. $\mathcal{S}$ is the number of samples.

Integrating Equation (17), (18), (19) and (23), the final loss function of our proposed model is:

$$\mathcal{L} = \mathcal{L}_{link} + \lambda_1 \mathcal{L}_{mat} + \lambda_2 \mathcal{L}_{cor} + \lambda_3 \mathcal{L}_{rec} \quad (24)$$

where $\lambda_1$, $\lambda_2$ and $\lambda_3$ are non-negative parameters.

## 3 Experiments

### 3.1 Experimental setup

*3.1.1 Datasets.* In this study, we utilise four widely-used datasets (*i.e.,* JF17K, WD50K, WikiPeople, and WikiPeople-) for evaluating our model's performance in handling link prediction tasks. JF17K is extracted and filtered from the Freebase knowledge base, while WD50K, WikiPeople and WikiPeople- are drawn from the Wikidata knowledge base. WikiPeople- is the variant of WikiPeople after dropping statements containing literals, only 2.6% facts contain

qualifiers. The arity of facts ranges from 2 to 7, making it a useful comparison against the original WikiPeople dataset for analyzing model performance on lower-arity facts. These datasets vary in terms of the number of facts, entities, relations, and the arity (number of arguments) they support[1].

*3.1.2 Comparison methods.* The comparison methods include nine intra-hyperrelational methods (*i.e.,* m-TransH [28], RAE[36], NaLP [7], HINGE [20], Transformer [6], STARE [6], ShrinkE [29], GRAN [26], and NYLON [35]) and two neighbourhood-aware methods (*i.e.,* HyperFormer [10] and HAHE [16]). m-TransH and RAE are generalised the existing TransH [27] model to handle multi-fold relations (*i.e.,* H-Facts) in HKGs directly. NaLP represents an H-Fact as a set of role-value pairs to capture the interactions between roles and values within an H-Fact. HINGE, Transformer, STARE, ShrinkE, GRAN, and NYLON represent an H-Fact as a main triple coupled with a set of qualifiers descriptive attribute-value pairs to directly learn the H-Fact representation in HKGs. HyperFormer and HAHE included entity neighbours to enhance the representations of H-Fact by capturing the structure information of HKGs.

*3.1.3 Setting-up.* All experiments are implemented in PyTorch and conducted on a server with two Nvidia V100 GPUs with 32GB of VRAM. We set $L = 2$, $T = 4$, and $d = 256$ for our hypergraph encoder, the learning rate as 3e-4, and set AdamW [14] as the optimizer. We reproduced the comparison methods according to the original paper to make them output the best results.

*3.1.4 Metrics.* We strictly follow the settings of (Galkin et al. [6], Luo et al. [16]) to predict a missing subject or object entity in a hyper-relational fact. We consider the widely used ranking-based metrics for link prediction, *i.e.,* mean reciprocal rank (MRR) and H@K (K=1,10).

---

[1]Relate work, the details of datasets, and hyperparameter settings can be found in Appendix.

Table 2: Link prediction performance of our method with different hypergraph on four datasets.

| $\mathcal{G}^{(s)}$ | $\mathcal{G}^{(r)}$ | $\mathcal{G}^{(o)}$ | $\mathcal{G}^{(av)}$ | JF17K | | | WD50K | | | WikiPeople- | | | WikiPeople | | |
|---|---|---|---|---|---|---|---|---|---|---|---|---|---|---|---|
| | | | | MRR | H@1 | H@10 | MRR | H@1 | H@10 | MRR | H@1 | H@10 | MRR | H@1 | H@10 |
| √ | | | | 0.402 | 0.342 | 0.515 | 0.333 | 0.264 | 0.465 | 0.261 | 0.180 | 0.428 | 0.396 | 0.290 | 0.568 |
| | √ | | | 0.655 | 0.621 | 0.722 | 0.464 | 0.443 | 0.503 | 0.503 | 0.481 | 0.543 | 0.529 | 0.502 | 0.577 |
| | | √ | | 0.369 | 0.298 | 0.502 | 0.248 | 0.191 | 0.356 | 0.246 | 0.179 | 0.386 | 0.373 | 0.278 | 0.521 |
| | | | √ | 0.541 | 0.461 | 0.701 | 0.268 | 0.200 | 0.397 | 0.236 | 0.155 | 0.407 | 0.418 | 0.336 | 0.552 |
| √ | √ | | | 0.603 | 0.560 | 0.682 | 0.479 | 0.443 | 0.545 | 0.540 | **0.507** | 0.601 | 0.516 | 0.475 | 0.591 |
| √ | | √ | | 0.304 | 0.247 | 0.409 | 0.337 | 0.275 | 0.457 | 0.271 | 0.202 | 0.405 | 0.414 | 0.314 | 0.570 |
| √ | | | √ | 0.455 | 0.401 | 0.556 | 0.332 | 0.263 | 0.465 | 0.267 | 0.183 | 0.441 | 0.409 | 0.302 | 0.582 |
| | √ | √ | | 0.605 | 0.566 | 0.680 | 0.435 | 0.411 | 0.478 | 0.505 | 0.483 | 0.543 | 0.517 | 0.495 | 0.557 |
| | √ | | √ | 0.687 | 0.656 | **0.748** | 0.479 | 0.460 | 0.512 | 0.510 | 0.491 | 0.543 | 0.515 | 0.496 | 0.547 |
| | | √ | √ | 0.418 | 0.347 | 0.556 | 0.242 | 0.185 | 0.350 | 0.248 | 0.180 | 0.382 | 0.371 | 0.276 | 0.518 |
| √ | √ | √ | | 0.434 | 0.379 | 0.543 | 0.476 | 0.442 | 0.541 | 0.527 | 0.494 | 0.590 | 0.532 | 0.494 | 0.604 |
| √ | √ | | √ | 0.641 | 0.606 | 0.708 | 0.445 | 0.406 | 0.520 | 0.532 | 0.496 | 0.604 | 0.515 | 0.470 | 0.596 |
| √ | | √ | √ | 0.300 | 0.239 | 0.412 | 0.328 | 0.267 | 0.446 | 0.281 | 0.204 | 0.439 | 0.407 | 0.305 | 0.567 |
| | √ | √ | √ | 0.614 | 0.570 | 0.700 | 0.442 | 0.419 | 0.484 | 0.503 | 0.480 | 0.542 | 0.519 | 0.493 | 0.567 |
| √ | √ | √ | √ | **0.696** | **0.669** | 0.745 | **0.488** | **0.453** | **0.553** | **0.544** | 0.506 | **0.614** | **0.538** | **0.499** | **0.611** |

## 3.2 Results and discussion

In this section, we present a detailed analysis of the results obtained from our experiments on four datasets in Table 1. First, our method achieves the best performance, followed by HyperFormer, HAHE, GRAN, ShrinkE, STARE, Transformer, NYLON, HINGE, NaLP, RAE, and m-TransH. For example, our method improves on average by 5.71% and 30.91%, respectively, compared to the best comparison method(*i.e.,* HyperFormer) and the worst comparison method (*i.e.,* m-TransH) in all cases. The results show the superior performance of our proposed method compared to various baselines. Second, compared with the intra-hyperrelational representation methods (*i.e.,* m-TransH, RAE, NaLP, HINGE, Transformer, STARE, ShrinkE, GRAN, and NYLON), has an average increase of 15.18%. This demonstrates the superiority of neighbourhood-aware representation methods, which may enhance representation learning and capture a more holistic and interconnected view of knowledge. Compared with the neighbourhood-aware methods (i.e., Hyper-Former, HAHE) has an average improvement of 6.14%. The results show that our method outperforms both intra-hyperrelational representation methods and existing neighbor-aware representation methods in capturing correlations in knowledge graphs. The most significant improvements are observed in H@1, where our model outperforms baselines averagely increased by 22.56%. This indicates that our model is more accurate at identifying the correct answer in the top position compared to the baselines, highlighting its effectiveness in the link prediction task.

Overall, our proposed method has two advantages. First, by constructing multi-hypergraphs, we can comprehensively utilise the neighbourhood information of H-Facts in HKGs. Second, by disentangling learning, we extract the common and private representations to exploit the semantic information of H-Facts and task-relevant information (*e.g.,* structure information of HKGs) of the HKRL.

## 3.3 Effectiveness on different hypergraph

Based on H-Fact's structural characteristics, we construct four different neighbourhood hypergraphs (*i.e.,* , subject hypergraph $\mathcal{G}^{(s)}$, relation hypergraph $\mathcal{G}^{(r)}$, object hypergraph $\mathcal{G}^{(o)}$, and qualifier hypergraph $\mathcal{G}^{(av)}$). In this section, we analyse the effectiveness of different combinations of these four hypergraphs, including single-hypergraph, two-hypergraph, three-hypergraph, and four-hypergraph combinations. The experimental results are shown in Table 2.

First, we evaluate the effectiveness of each hypergraph individually. Compared with other hypergraphs, the relation hypergraph are 14.99% higher on average, indicating that the similarity of relation in HKG promotes representation learning and is conducive to mining semantic information of H-Fact. Second, the combination of relation hypergraph and qualifier hypergraph improves by an average of 12.58% compared with other variants. This result shows that relation similarity and qualifier identity in HKG are crucial to mining comprehensive semantic patterns and improving hyper-relational knowledge representation. Third, the method of combining four hypergraphs achieves the best effect, which indicates that the construction of multi-hypergraph for hyper-relational knowledge representation is effective and comprehensively utilises the neighbourhood information of H-Facts.

## 3.4 Ablation study

The proposed MHD has four loss functions, *i.e.,* a matching loss $\mathcal{L}_{mat}$ (see Eq.(17)), correlation loss $\mathcal{L}_{cor}$ (see Eq.(18)), reconstructed loss $\mathcal{L}_{rec}$ (see Eq.(19)) and link prediction loss $\mathcal{L}_{link}$ (see Eq.(23)). To demonstrate the effectiveness of each part, we tested different combinations of these loss functions (except $\mathcal{L}_{link}$ as we cannot perform link prediction tasks without $\mathcal{L}_{link}$) on the link prediction task by reporting the results in Table 3.

Table 3: Ablation study of our method on four datasets.

| $\mathcal{L}_{link}$ | $\mathcal{L}_{mat}$ | $\mathcal{L}_{cor}$ | $\mathcal{L}_{rec}$ | JF17K | | | WD50K | | | WikiPeople- | | | WikiPeople | | |
|---|---|---|---|---|---|---|---|---|---|---|---|---|---|---|---|
| | | | | MRR | H@1 | H@10 | MRR | H@1 | H@10 | MRR | H@1 | H@10 | MRR | H@1 | H@10 |
| √ | | | | 0.354 | 0.301 | 0.453 | 0.431 | 0.388 | 0.512 | 0.514 | 0.481 | 0.577 | 0.508 | 0.454 | 0.604 |
| √ | √ | | | 0.602 | 0.561 | 0.678 | 0.446 | 0.405 | 0.524 | 0.526 | 0.489 | 0.595 | 0.512 | 0.478 | 0.576 |
| √ | | √ | | 0.684 | 0.653 | 0.739 | 0.448 | 0.406 | 0.526 | 0.507 | 0.468 | 0.583 | 0.514 | 0.473 | 0.592 |
| √ | √ | √ | | 0.684 | 0.652 | 0.744 | 0.440 | 0.398 | 0.518 | 0.526 | 0.491 | 0.593 | 0.528 | 0.489 | 0.601 |
| √ | | | √ | 0.301 | 0.244 | 0.404 | 0.460 | 0.419 | 0.536 | 0.496 | 0.461 | 0.560 | 0.516 | 0.473 | 0.597 |
| √ | √ | | √ | 0.685 | 0.657 | 0.738 | 0.484 | 0.451 | 0.547 | 0.502 | 0.462 | 0.581 | 0.525 | 0.485 | 0.598 |
| √ | | √ | √ | 0.688 | 0.664 | 0.733 | 0.483 | 0.445 | **0.554** | 0.517 | 0.482 | 0.585 | 0.533 | 0.491 | 0.609 |
| √ | √ | √ | √ | **0.696** | **0.669** | **0.745** | **0.488** | **0.453** | 0.553 | **0.544** | **0.506** | **0.614** | **0.538** | **0.499** | **0.611** |

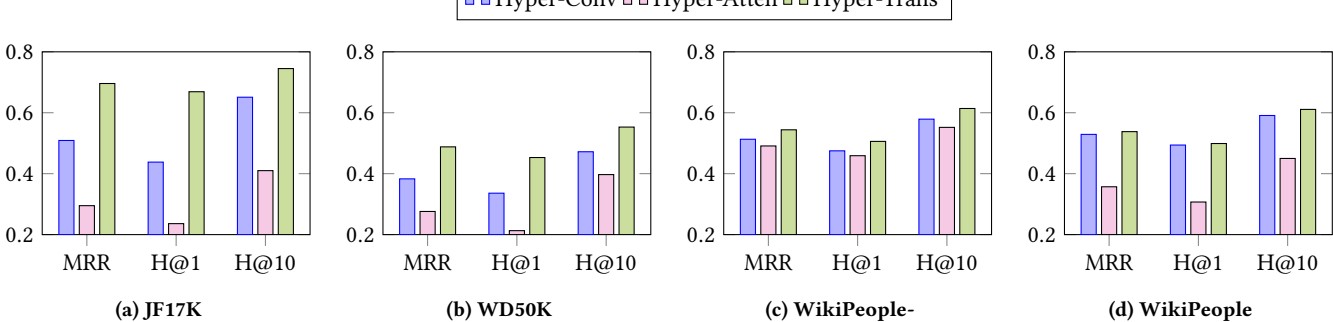

(a) JF17K  (b) WD50K  (c) WikiPeople-  (d) WikiPeople

Figure 3: Effectiveness of different hypergraph encoder on four datasets

First, we remove the disentangled learning module (*i.e.,* without $\mathcal{L}_{mat}$, without $\mathcal{L}_{cor}$ and without $\mathcal{L}_{rec}$) to evaluate multi-hypergraph method for HKRL. The MHD has an average increase of 14.46% compared to the variant (*i.e.,* without disentangled learning module). The result indicated that multi-hypergraph disentanglement can improve the ability of HKRL. Second, with the addition of $\mathcal{L}_{rec}$, these variants grew by an average of 1.34%. This shows that reconstruction loss can effectively improve the model. Third, the best performance is achieved by our method, which utilises the entire loss function. Thus, our method's different loss functions have distinct contributions that are in accordance with our motivation.

## 3.5 Effectiveness of different encoder

To verify the effectiveness of our proposed hypergraph transformer encoder, we conducted comparative experiments on link prediction by replacing the encoder module across four datasets. Specifically, we compared the hypergraph convolution encoder (*i.e.,* Hyper-Conv), the hypergraph attention encoder (*i.e.,* Hyper-Atten), and the hypergraph transformer encoder (*i.e.,* Hyper-Trans). The detailed experiments are shown in Figure 3. Compared to the hypergraph convolution method, the hypergraph attention encoder achieved an average improvement of 9.74%. Compared to the hypergraph attention method, the hypergraph attention encoder achieved an average improvement of 21.23%. The experiments demonstrate that our customised hypergraph transformer encoder is more effective in extracting features from H-Facts for HKRL. It can capture the relationships between any two nodes in the hypergraph and learn the intrinsic connections and importance differences between nodes through the self-attention mechanism.

## 4 Conclusion

In this paper, we have presented a novel multi-hypergraph disentanglement method (MHD) for hyper-relational knowledge representation learning. Specifically, we constructed multiple hypergraphs to mine the relation similarity and qualifier identity in HKGs and comprehensively utilise the neighbourhood information of H-Facts. In addition, we used disentangle representation learning to mine clean semantic information of H-Fact and complete structure information of HKG. To the best of our knowledge, this is the first attempt to decouple the common and private representations of multi-hypergraphs by disentangling learning. Finally, the fusion representation is used to conduct a link prediction task. Extensive experimental results demonstrate that the proposed MHD method consistently achieves state-of-the-art performance in hyper-relational knowledge representation learning.

In future work, we would like to explore more intelligent techniques for constructing hypergraphs, *e.g.,* dynamic hypergraph construction, potentially incorporating deep learning architectures to automatically discover and represent relation similarities and qualifier identities within HKGs. Additionally, we intend to extend the application of our MHD method to various hyper-relational knowledge representation tasks, including entity classification, relation extraction, and knowledge reasoning. This expansion will enable us to confirm its adaptability and broad applicability.

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

# A Related work

This section briefly reviews the work of hyper-relational knowledge representation learning. According to the criteria of whether to use neighbourhood information, we divided the existing work into two categories, *i.e.,* intra-hyperrelational representation methods and neighbourhood-aware representation methods.

## A.1 Intra-hyperrelational representation methods

Intra-hyperrelational representation methods extract the representation of each element by learning the relationships between the elements of H-Fact [1, 23]. For example, in 2016, Wen et al. [28] argued that existing models, which convert multi-fold (or H-Fact) relational data into binary triples, result in a loss of structural information and introduce heterogeneity in predicates, making representation less effective. Thus, the authors generalize the well-known TransH model to create a new model called m-TransH by capturing the roles of entities in multi-fold relations. Based on this work, Zhang et al. [36] presented RAE model enhances HKGs representation techniques by incorporating a relatedness metric that captures the likelihood of entities co-participating in a common H-Fact. In addition, Guan et al. [7] transferred an H-Fact into a set of role-value pairs and introduced the NaLP model to capture the interdependencies between different roles and their corresponding values within the same H-Fact. These methods enhance the representation of traditional knowledge graphs by computing individual H-Fact features. However, they ignore the information provided by the attribute element in the qualifier of H-Fact.

Rosso et al. [20] address this by proposing HINGE, which learns from H-Facts, where each fact is not just a simple triplet but also includes additional key-value pairs that offer further contextual information. The model uses convolutional neural networks to learn from the base triplet $(s, r, o)$ and the additional attribute-value pairs $(a, v)$. In 2020, Galkin et al. [6] presents a graph encoder called STARE. The model is based on graph neural networks (GNNs), which use a message-passing mechanism that learns representations for both the main triplet and any additional qualifiers. They discuss how STARE can preserve the role of both main and qualifying information in H-Facts, a first for GNN-based approaches. Inspired by the prosperity in GNNs, Wang et al. [26] proposes the GRAN model, which represents each H-Fact as a heterogeneous graph. This graph includes entities, relations, and attributes as nodes and uses four types of edges (*i.e.,* subject-relation, object-relation, relation-attribute, attribute-value) to capture the interactions between nodes. GRAN uses edge-biased fully-connected attention to learn from these heterogeneous graphs. This attention mechanism is designed to handle both local and global dependencies within the H-Fact.

However, in practical scenarios, H-Facts are often interconnected rather than isolated. These existing works ignore the relationships between H-Facts and the structure information of HKGs.

## A.2 Neighbourhood-aware representation methods

The neighbourhood-aware representation method was proposed to address the limitations of the intra-hyperrelational representation method. This approach incorporates the neighbourhood features of an H-Fact to enhance the representation learning process and capture a more holistic and interconnected view of knowledge. For example, Hu et al. [10] designed a HyperFormer model with entity neighbor aggregator to addresses challenges in previous models that rely on graph structures by integrating information from an entity's one-hop neighbors. Luo et al. [16] proposed the HAHE, which models both the global hypergraph structure and the local sequential structure of H-Facts. The global-level hypergraph is built by representing entities (*i.e., s, o,* and *v*) as nodes and H-Facts as hyperedges. The relationship between nodes and hyperedges is captured by an incidence matrix, and dual-attention layers propagate information between nodes and hyperedges, allowing the HAHE model to capture the global structure of the HKG.

However, their methods only focus on direct neighbourhood information of entities (*i.e., s, o,* and *v*), neglecting the relation (*i.e., r*) similarity of the main triple and the attribute (*i.e., a*) identity in the qualifiers. Secondly, existing approaches often construct a single perspective to mine the neighbourhood information of H-Facts. For example, HAHE [16] constructed a hypergraph to represent the connectivity of entities and used hypergraph dual-attention layers to capture topological relationships between entities. These methods ignore HKG's diversity, decreasing the quality of the learned representations. Therefore, combining different perspectives to construct multi-hypergraph can capture the diversity of the HKG to mining comprehensive semantic information and improving the representation of H-Facts.

# B Datasets

In this paper, we evaluate the performance of our method using four benchmark datasets, *i.e.,* JF17K, WD50K, WikiPeople, and WikiPeople-. Table 4 provides a summary of the statistics for these datasets. Each dataset contains hyper-relational facts (H-facts) that are expressed as a main triple (subject, relation, object) supplemented by a variable number of qualifiers (attribute-value). The datasets differ in their scale, the proportion of H-Facts, the number of entities, relations, and the arity (the number of elements in each fact, including qualifiers). We list the details of the datasets as follows:

- **JF17K**[2] contains 100,947 facts, with 45.9% being hyper-relational facts (46,320 facts). The dataset includes 28,645 unique entities and 501 relations. It is divided into 76,379 training facts and 24,568 testing facts. Since there is no validation set for JF17K, we randomly sample 20% of the train set as a validation set.
- **WD50K**[3] contains a total of 236,507 facts, with 13.6% hyper-relational facts (32,167 facts). The dataset includes 47,156 entities and 532 relations, split into 166,435 training facts,

---

[2]https://github.com/lijp12/SIR
[3]https://zenodo.org/record/4036498

### Table 4: Statistics of all datasets

| Dataset | All facts | Hyper-relational facts(%) | Entities | Relations | Train | Valid | Test | Arity |
|---|---|---|---|---|---|---|---|---|
| JF17K | 100,947 | 46,320(45.9%) | 28,645 | 501 | 76,379 | - | 24,568 | 2-6 |
| WD50K | 236,507 | 32,167(13.6%) | 47,156 | 532 | 166,435 | 23,913 | 46,159 | 2-67 |
| WikiPeople | 382,229 | 44,315(11.6%) | 47,765 | 193 | 305,725 | 38,223 | 38,281 | 2-9 |
| WikiPeople- | 369,866 | 9,482(2.6%) | 34,825 | 178 | 394,439 | 37,715 | 37,712 | 2-7 |

### Table 5: Settings for the proposed MHD

| Settings | JF17K | WD50K | WikiPeople- | Wikipeople |
|---|---|---|---|---|
| $L$ | 2 | 2 | 2 | 2 |
| $T$ | 4 | 4 | 4 | 4 |
| $d$ | 256 | 256 | 256 | 256 |
| $d_c$ | 192 | 192 | 192 | 192 |
| $d_p$ | 64 | 64 | 64 | 64 |
| $\lambda_1$ | 1e-6 | 1e-4 | 1e-5 | 1e-5 |
| $\lambda_2$ | 1 | 1 | 1 | 1 |
| $\lambda_3$ | 1 | 1 | 1 | 1 |
| learning rate | 0.0001 | 0.0003 | 0.0003 | 0.0003 |
| batch size | 1024 | 128 | 1024 | 512 |
| epoch | 100 | 100 | 100 | 200 |
| weight decay | 0.01 | 0.01 | 0.01 | 0.01 |
| dropout | 0.1 | 0.1 | 0.1 | 0.1 |

23,913 validation facts, and 46,159 test facts. The arity of
the facts ranges from 2 to 67.

- **WikiPeople**[4] comprises 382,229 facts, of which 11.6% are
  hyper-relational (44,315 facts). It includes 47,765 entities
  and 193 relations, divided into 305,725 training facts, 38,223
  validation facts, and 38,281 test facts. The arity of the facts
  ranges from 2 to 9.
- **WikiPeople-** is a variant of the Wikipeople dataset that
  filters out statements containing literals with a lower pro-
  portion of hyper-relational facts (2.6%). The dataset con-
  tains 34,825 entities and 178 relations, with 394,439 training
  facts, 37,715 validation facts, and 37,712 test facts. The arity
  of the facts ranges from 2 to 7.

## C  Hyperparameter settings

In this paper, we train the MHD method on two 32G V100 GPUs.
Table 5 describes the detailed settings of our experimental setups
with MHD. MHD takes approximately 2 hours to complete the
training and evaluation on JF17K, 10 hours on WD50K, 6 hours on
WikiPeople-, and 8 hours on WikiPeople.

---

[4]https://github.com/gsp2014/NaLP

