# OpenReview forum: "Hyper-Relational Knowledge Representation Learning with Multi-Hypergraph Disentanglement"
_ACM.org/TheWebConf/2025/Conference — WWW 2025 Poster_

### Official Review · Reviewer_advK · 2024-11-07

**Novelty:** 3
**Technical Quality:** 4

**Review:**

The paper presents an approach to hyper-relational knowledge representation learning (HKRL) by introducing a multi-hypergraph disentanglement method (MHD). The method aims to address the limitations of existing HKRL methods by capturing relational similarity and attribute identity within hyper-relational facts (H-Facts) and utilizing neighborhood information effectively.

**Questions:**

Pros:

1. The use of disentangled representation learning to separate semantic information from structural information is a sophisticated approach that can enhance the interpretability of the learned representations.

2. The paper provides a comprehensive experimental evaluation of four real-world datasets, demonstrating the effectiveness of the proposed method over state-of-the-art baselines.

Cons:

1. While the paper claims the method's effectiveness, it lacks a deeper analysis of why the multi-hypergraph approach outperforms single-graph methods, especially in terms of theoretical insights.

2. Perform ablation studies to isolate the effects of the multi-hypergraph construction and disentangled learning components on the performance of the MHD method.

3. Measure the training time, memory usage, and scalability concerning the number of entities, relations, and hyper-relational facts.

4. While the paper mentions the interpretability of disentangled representations, there is a lack of experiments or case studies that demonstrate how the disentangled representations can be used to explain predictions or provide insights into the knowledge graph structure.

5. The paper does not sufficiently explore how different types and numbers of qualifier attributes affect the performance of the MHD method.


Questions:

1. The paper mentions that the proposed method can handle noise and interference better than previous methods. Provide more details about how the disentangled learning and the multi-hypergraph construction specifically contribute to noise reduction. Are there any experiments or analyses to support this claim?

2. In the link prediction task, the attention mechanism is used to fuse the private representations. Explain how the attention mechanism is trained and what factors influence its ability to dynamically adjust the contribution of each hypergraph. Are there any potential drawbacks or limitations to using this attention mechanism?

3. In the multi-hypergraph construction process, how do you ensure the balance between the different hypergraphs?

**Reviewer Confidence:**

3: The reviewer is confident but not certain that the evaluation is correct

**Scope:**

3: The work is somewhat relevant to the Web and to the track, and is of narrow interest to a sub-community

---

### Official Review · Reviewer_dwr3 · 2024-11-22

**Novelty:** 7
**Technical Quality:** 6

**Review:**

The paper introduces a novel learning approach for hyper-graphs learning that incorporates attributes and multi-layer representations.

The approach is validated through extensive experiments on four datasets, outperforming the existing approaches. The work is detailed comparisons against state-of-the-art methods and include an ablation study that demonstrate the effectiveness of each loss function. Besides, the effectiveness of the encoder is validated.

The presentation is clear and comprehensive. The dataset description and the training details are presented.
The work is original in addressing hyper-graph learning challenges. Its application to the linking task makes it a significant contribution to the field.

Pros:

Strong Experimental Validation.

Well-Designed Ablation Study.

Novel Methodology.

Cons:

Missing comparison with other embedding-based method for link prediction.

Not considering other evaluation metrics like precision and recall allowing to better understand the model's performance.

**Questions:**

Why is there no comparison with other embedding-based methods for designed to perform link prediction like BERT-INT[1]?

Why were precision and recall excluded from the evaluation metrics?



[1]: BERT-INT:A BERT-based Interaction Model For Knowledge Graph Alignment, Xiaobin Tang, Jing Zhang, Bo Chen, Yang Yang, Hong Chen, Cuiping Li, IJCAI 2020.

**Reviewer Confidence:**

3: The reviewer is confident but not certain that the evaluation is correct

**Scope:**

4: The work is relevant to the Web and to the track, and is of broad interest to the community

---

### Official Review · Reviewer_J2oH · 2024-11-25

**Novelty:** 3
**Technical Quality:** 3

**Review:**

This paper proposes a multi-hypergraph disentanglement method for hyper-relational knowledge representation learning, aiming to address limitations in existing methods by mining common and private information across four constructed hypergraphs. Experimental results on four datasets demonstrate the method's effectiveness, achieving state-of-the-art performance on link prediction tasks for hyper-relational knowledge graphs.

**Strengths:**

1. The paper is well-organized, with clear explanations of the distinctions and advantages of the proposed method compared to existing works.

2. The release of source code enhances reproducibility and facilitates future research in HKRL.

3. The experiments are extensive, providing performance comparisons on multiple datasets and demonstrating competitive results.

**Drawbacks:**

1. The method relies heavily on an ensemble of four hypergraphs and four kinds of losses, resulting in computational redundancy. While the combination of these elements yields optimal results, the ensemble nature offers limited research insights or theoretical advancements.

2. The paper bases its approach on intuitive assumptions without theoretical analysis or preliminary evidence. For instance, the claim that “extract common and private information across multiple views to minimize noise and interference” lacks empirical or theoretical backing till the end. Whether the performance improvement is due to this effect has not been confirmed.

3. Several advantages of the proposed attention and ensemble mechanisms are exaggerated without sufficient justification:

+ Lines 460-462: The claim that the self-attention mechanism mitigates missing data and noise is contradicted by significantly worse performance when using single hypergraphs, which raises questions about the robustness of the method.

+ Lines 572-576: The proposed “attention” function, implemented as a basic MLP with concatenated input features, does not convincingly address the shortcomings of summation or average pooling. Results from Table 2 show that two- or three-hypergraph configurations often underperform single-hypergraph models, challenging the motivation.

4. The performance discrepancy between single-hypergraph variants and the complete MHD model is not well-explained. For instance, relation hypergraphs outperform other individual variants and closely approach MHD performance, which undermines the justification for including other hypergraphs.

5. The paper does not discuss model complexity, training parameters, or efficiency compared to baseline methods. The ensemble design and use of four hypergraphs imply high computational cost, which is a critical omission.

6. The authors suggest that combining hypergraphs into one may dilute or confuse key features, However, some multi-hypergraph variants of the proposed method also experience a performance decrease. Only the full model significantly outperforms all other variants. This contradiction is not addressed or analyzed in the paper.

**Questions:**

1. Does the hypergraph encoder share trainable parameters across the four hypergraph types? If not, how is parameter efficiency ensured?

2. How are the losses from different hypergraphs combined? Are they simply summed, or is there a specific weighting scheme?

**Reviewer Confidence:**

3: The reviewer is confident but not certain that the evaluation is correct

**Scope:**

3: The work is somewhat relevant to the Web and to the track, and is of narrow interest to a sub-community

---

### Official Review · Reviewer_RxFS · 2024-11-30

**Novelty:** 6
**Technical Quality:** 5

**Review:**

## Short Summary
In this work, the authors introduce a Multi-Hypergraph Disentanglement (MHD) method for hyper-relational knowledge representation learning.

## Definition/Notation
As per my understanding of your work, literals are not considered. If this is the case, the paper should explicitly state that qualifiers taking literals as values are excluded. To address this, Section 2 should be revised to clearly define the exact nature of the values considered. Additionally, referring to qualifiers as "auxiliary attributes" in some parts of the text can be confusing—ensure consistent terminology throughout the paper.

## Limitations
Further directions are provided for future work. However, the limitations of the approach are not explicitly discussed. Add a dedicated section to discuss the limitations of your approach.

## Discussion of Results
The results seem promising. However, the discussion of results is insufficient. Comparing the obtained results based on the characteristics of the datasets. (E.g., model performance vs. sparcity of a dataset)

While the reported results are promising, the discussion lacks depth. Consider providing a more comprehensive analysis of the results by comparing the model's performance based on dataset characteristics, such as sparsity.

## Results
There is a mistake in Table 2. The best Hits@1 result on dataset WD50K is not 0.453 from the last row but 0.460 from the 9th row.


## Minor comments:
1. In Section 1,  on line 111, there is a typo - “... , so that they difficult extract …”.
2. You have “The arity of facts ranges from 2 to 7 .. “,  “Arity (number of arguments) … ”, and “arity (the number of elements in each fact, including qualifiers)” on lines 660,663, and 1145. You should define it clearly the first time you use it.
3. There is a typo on line 695

**Questions:**

1. Can your approach be generalized to account for literals as well?
2. On line 420, you mention using either word embeddings or random initialization. Which option performs better, and on which datasets?
3. Is your method adaptable for inductive link prediction (LP)?
4. How effectively does your approach generalize across diverse datasets? Does it scale efficiently?
5. What are the primary limitations of your method?

**Reviewer Confidence:**

4: The reviewer is certain that the evaluation is correct and very familiar with the relevant literature

**Scope:**

4: The work is relevant to the Web and to the track, and is of broad interest to the community

---

### Official Review · Reviewer_gHPK · 2024-12-01

**Novelty:** 5
**Technical Quality:** 5

**Review:**

The paper proposes multi-hypergraph disentanglement method for hyper-relational knowledge representation learning, which mines the relation similarity and disentangles node representation into common and pravite representations.
1. Authors seperately consider the feature of entity, the relation of nodes and the attribute to construct multiple different hypergraph, which mines comprehensive semantic information in HKGs.
2. Authors think that using the mutli-head attention to learn the hypergraph representations can mitigate the impact of noise on model performance. Can you demonstrate through visualization that noise is reduced？
3. In Section 3.5, the authors provide a comparison with other methods for constructing hypergraphs. However, what are the differences between the hypergraph attention encoder and the hypergraph transformer encoder? Why does the performance of the former appear to be much poor?
4. The use of a disentangled module to separate common and private information across views is commo. How does the decoupling method used in this work differ from other approaches?
5. In Section 3.3, using only $\mathcal{G}^{(r)}$ appears to yield promising results, while adding other hypergraphs actually reduces performance. Could the authors provide a reasonable explanation for this observation?

**Questions:**

1. Authors think that using the mutli-head attention to learn the hypergraph representations can mitigate the impact of noise on model performance. Can you demonstrate through visualization that noise is reduced？
2. In Section 3.5, the authors provide a comparison with other methods for constructing hypergraphs. However, what are the differences between the hypergraph attention encoder and the hypergraph transformer encoder? Why does the performance of the former appear to be much poor?
3. The use of a disentangled module to separate common and private information across views is commo. How does the decoupling method used in this work differ from other approaches?
4. In Section 3.3, using only $\mathcal{G}^{(r)}$ appears to yield promising results, while adding other hypergraphs actually reduces performance. Could the authors provide a reasonable explanation for this observation?

**Reviewer Confidence:**

4: The reviewer is certain that the evaluation is correct and very familiar with the relevant literature

**Scope:**

4: The work is relevant to the Web and to the track, and is of broad interest to the community